# Analysis of Factors Influencing the Spatial and Temporal Variability of Carbon Intensity in Western China

**Mingchen Yang, Lei Wang * and Hang Hu**

College of Water Conservancy & Architectural Engineering, Shihezi University, Shihezi 832000, China;
20222010006@stu.shzu.edu.cn (M.Y.); 20222110039@stu.shzu.edu.cn (H.H.)
* Correspondence: wl1128@shzu.edu.cn

**Abstract:** Understanding the spatio-temporal differentiation of carbon intensity factors is crucial for setting scientific and reasonable carbon emission reduction targets. This study, based on relevant data from the western regions for the years 2010–2019, analyzes the influencing factors of the spatio-temporal distribution differences in carbon intensity in these areas. Additionally, the Grey Forecasting Model was utilized to predict the development trend of average carbon intensity in the western regions. The results indicate the following: (1) The temporal dimension of carbon intensity in the western regions shows an overall declining trend with local rebounds, while the high-value areas of spatial carbon intensity are concentrated in the northern part of the study area. (2) Per capita Gross Domestic Product, energy consumption per unit of Gross Domestic Product and investment in industrial pollution control have a positive impact on carbon intensity, whereas investment in the energy industry and per capita disposable income of residents have a negative impact. (3) Energy consumption per unit of Gross Domestic Product is the factor with the highest degree of explanation in univariate analysis; interaction detection results suggest that the core factors of spatial distribution differences in carbon intensity are energy consumption and urban development. (4) Predictions using the Grey Forecasting Model for the development of carbon intensity in the western regions show a year-by-year decline, consistent with carbon intensity control targets. Based on these conclusions, this paper proposes policy recommendations focusing on improving regional economic coordination mechanisms, increasing investment in industrial pollution control, managing energy industry expenditures, adjusting the proportion of the urban population, and enhancing the per capita disposable income of residents.

**Keywords:** western region; carbon intensity; STIRPAT model; Geo-detector model; GM (1:1) model

## 1. Introduction

Climate change is primarily driven by the release of greenhouse gases, with carbon dioxide being a major contributor [1,2]. The consequences of climate change, including elevated global temperatures, heightened frequency of extreme weather events, and rising sea levels, are increasingly posing a substantial threat to global society, the economy, and the environment. The international community has been proactively responding to the challenges posed by climate change. The Kyoto Protocol demarcated that a global effort in climate change mitigation aims to reduce GHG emissions from countries (developed countries) by 5.2%, with the year 1990 as baseline in the first commitment period of 2008–2012 [3]. The Glasgow Climate Pact underscored the necessity for swift, substantial, and enduring diminutions in worldwide greenhouse gas emissions to cap global warming at 1.5 °C [4]. The long-run goal of the Paris Agreement is to maintain the global mean temperature increase below 2 °C over the pre-industrial period and to pursue an effort to limit the temperature rise to 1.5 °C [5]. The shift from the Kyoto Protocol to the Paris Agreement signifies a global acknowledgment of the severity of the climate change issue. The coming into force of the Paris Agreement has ushered in a new dawn for global cooperation on climate change [6].

China has taken proactive measures to address the global climate change challenge. The country has made commitments to peak carbon dioxide emissions by 2030 and achieve carbon neutrality by 2060 in alignment with the global response to climate change [7,8]. In light of this, China has adopted the carbon intensity control target as the low-carbon-emission reduction objective, aligning it with the country's specific circumstances. This approach ensures the fulfillment of low-carbon responsibilities while concurrently safeguarding the progress of economic development. Carbon intensity represents the amount of carbon dioxide emissions per unit of GDP (Gross Domestic Product) and is influenced by the interplay between the economy and carbon emissions [9,10]. It serves as a crucial metric for assessing both economic development and progress toward low-carbon initiatives. The extensive area and notable carbon emissions linked to economic activities position China's western region as a pivotal area for economic and environmental research. To discern the key determinants affecting the disparities in carbon intensity distribution within these regions, this study harnesses regional data, positing carbon intensity as the dependent variable. A selection of independent variables known to influence carbon intensity was methodically analyzed. The STIRPAT (Stochastic Impacts by Regression on Population, Affluence, and Technology) model was utilized to assess factors affecting the temporal variation of carbon intensity, while the Geo-detector (Geo-graphy detector) model was deployed to explore the spatial determinants. Finally, a comprehensive analysis of these factors was conducted, and the GM (1,1) model (Gray Forecast Model) was employed to forecast the development of carbon intensity in the western regions. Focusing on the specifics of carbon intensity management in the western regions paves the way for the development of informed policies. This detailed examination improves the distribution of resources, directly influencing the practical application of sustainable solutions. Such an approach not only propels these regions towards sustainable development but also ensures that strategies are grounded in real-world applications, fostering environmental sustainability alongside economic growth.

## 2. Literature Review

Current research on the spatial and temporal differentiation of carbon intensity primarily involves scholars analyzing the spatial and temporal distribution of carbon intensity and exploring the factors influencing the differences in the spatial and temporal distribution. In analyzing the spatial and temporal distribution of carbon intensity, Congqi Wang and Pengzhen Liu conducted a path analysis of the spatial and temporal evolution of green finance and carbon emissions in the Pearl River Delta (PRD) region, and investigated the direction of the spatial evolution of green finance and carbon emissions and the spatial spillover effect of carbon emissions [11]. Lang Xu and Zhihui Yang analyzed the spatial and temporal evolution characteristics and spillover effects of carbon emissions from shipping trade in EU coastal countries, and found that carbon emissions from shipping trade in EU coastal countries have positive spatial correlation and spatial clustering [12]. Xiaoyan Sun calculated the power generation intensity of 30 provinces, and analyzed the spatial and temporal characteristics of power generation carbon emission intensity and the spatial spillover effects of the drivers in each province in China [13]. Ying Zhou and Dan Hu identified the main influencing factors and calculated their impacts using the LMDI model. They then explored the decoupling relationship between carbon emissions and economic output using the Tapio decoupling index. Finally, they analyzed the temporal and spatial evolution of carbon emissions through spatial autocorrelation theory [14]. In the study of Lei Li and Junfeng Li, an exploratory spatial data analysis (ESDA) framework was constructed through spatial autocorrelation, kernel density estimation and standard deviation ellipse to analyze the spatio-temporal evolutionary characteristics of carbon emissions in the Greater Bay Area, and to identify various influencing factors of carbon emissions in the Greater Bay Area by combining geographically and temporally weighted regression (GTWR) models [15]. Fuqiang Han and Alimujiang Kasimu et al. used the arid regions of Northwest China as their research subject and analyzed the carbon balance

under land use changes by combining top-down and bottom-up approaches [16]. Suwen Xiong and Fan Yang analyzed the carbon balance patterns of city clusters in the central Yangtze River region. Combined with comprehensive trend predictions, they proposed a mechanism for regulating carbon balance [17]. Yao Zhang and Jing Quan used the city of Xi'an as their research subject to construct a model for estimating regional carbon emission fines based on energy consumption and NPP-VIIRS type NTL data, and quantitatively analyzed multi-scale carbon emissions from 2000 to 2021 [18]. Lanyi Zhang and Dawei Weng studied the provincial road transportation carbon emissions in China from 2006 to 2021, finding that China's road transportation carbon emissions exhibit an east-high and west-low distribution. They analyzed five factors influencing transportation carbon emissions [19]. Lijuan Su and Yatao Wang measured the spatial and temporal evolution characteristics and convergence of agricultural carbon emissions, conducted a comparative analysis of regional differences, and investigated the spatial correlation and spatial spillover effects using panel data from 31 provinces in China from 2005 to 2020 [20].

In exploring the factors influencing the differences in the spatial and temporal distribution, some scholars have consolidated insights from previous studies on the characteristics of spatio-temporal variations in carbon intensity. They analyzed these variations using various influential factor analysis models. Kaile Zhou and Jingna Yan analyzed the spatio-temporal evolution characteristics and spillover effects of regional carbon emissions in China. They found that the level of regional technological innovation dampens carbon emissions and that an increase in neighboring regions will similarly dampen local carbon emissions [21]. Shengnan Cui and Yanqiu Wang analyzed the multifactorial spatio-temporal carbon emissions in China from a holistic governance perspective, constructing a spatio-temporal decomposition method and a two-dimensional separation model [22]. Yi Yang and Huan Qin investigated the spatio-temporal and regional heterogeneity of carbon peak uncertainty and its triggers in China, discovering that the carbon peak depends on the rate of change in carbon intensity and per capita GDP. They also found that the drivers for carbon decoupling vary across provinces [23]. Siying Chen studied 108 cities in the Yangtze River Economic Belt and used a combined coordination degree model and an optimal parameter-based geoprobe model to assess the synergy level between pollution control and carbon reduction, as well as to identify its driving factors [24]. Xiaoying Liang and Min Fan used 30 provinces in China as their research subjects to calculate the spatial and temporal distribution characteristics of energy carbon emissions using the carbon emission coefficient method. They also analyzed the driving factors behind the differences in the spatial and temporal distribution characteristics [25]. Hao Lu, Chengyou Xiao, and Liudan Jiao analyzed the impact mechanism from three perspectives: vehicle intelligence, road intelligence, and cloud data. They empirically analyzed the spatial differences in carbon emission control based on panel data from 285 cities in China [26]. Miao He expanded the traditional STIRPAT framework to examine the impact of market integration on carbon emission coefficients and heterogeneity in Eastern China, as well as to identify factors influencing carbon intensity and low-carbon pathways in market development [27]. Xiaoyi Shi and Xiaoxia Huang employed Social Network Analysis (SNA) to study the spatial correlation network of China's carbon emissions (CCESCN) from 2011 to 2020, analyzing the factors influencing the evolution characteristics of carbon emissions [28].

The field of low-carbon development, particularly in studies of carbon intensity, focuses primarily on two key areas. First, it includes the analysis of carbon intensity's spatial and temporal disparities. Second, it involves investigating the current spatial and temporal variations in factors that influence carbon intensity. Understanding both the spatial–temporal characteristics of carbon intensity and the factors that affect it is crucial for crafting effective carbon emission reduction policies and strategies. This understanding not only aids in the development of scientifically based reduction measures but also offers valuable insights for fostering inter-regional collaboration in reducing carbon emissions. Despite the comprehensive nature of existing research on the spatial and temporal differences in carbon intensity and the analysis of its influencing factors, our literature review

indicates a strong scholarly emphasis on exploring the factors influencing spatial differences. Some researchers have adopted an analysis method focusing on influencing factors across two dimensions to study carbon intensity disparities. However, this method may not sufficiently highlight the dominant role of influencing factors within a specific dimension. Building on previous studies, this paper analyzes the influencing factors associated with spatial and temporal differences in carbon intensity, considering both dimensions to provide a more nuanced understanding.

## 3. Research Design

### 3.1. Variable Selection

3.1.1. Selection of the Dependent Variable

Carbon intensity signifies the interplay between carbon emissions and economic growth, serving as a crucial indicator for assessing the extent of carbon peaking in a city. Carbon intensity (Y) is determined by dividing carbon emissions by GDP [9,10,29], as in this study. This study focuses on the western regions of China, excluding Tibet due to its unique geographical and socio-economic challenges that complicate data collection and analysis, potentially affecting data accuracy and comparability. The research aims to identify and analyze trends and factors relevant across the western regions, excluding areas like Tibet that require distinct, specialized analytical methods. Thus, Tibet's exclusion does not impact the overall research findings, ensuring a focused and applicable analysis for the targeted regions.

3.1.2. Choice of Independent Variables

Given that carbon intensity serves as a comprehensive indicator derived from carbon emissions and GDP, the choice of independent variables for assessing carbon intensity should encompass factors related to both carbon emissions and GDP. The independent variables for carbon intensity include energy consumption, urban development level, economic development level, and pollution control. In the context of carbon emissions, the dynamics of carbon intensity are shaped by the interplay of energy consumption and pollution control. Energy consumption comprises elements like investment in the energy industry, overall energy consumption, and energy consumption per unit of GDP. Meanwhile, pollution control encompasses investments directed at industrial pollution control measures. The carbon intensity is influenced by both the degree of urban development and economic advancement, with a focus on GDP. The urban development level encompasses factors such as the proportion of the urban population, per capita disposable income, and the total regional population. On the other hand, the economic development level involves per capita GDP and the proportion of the secondary and tertiary industries in the GDP composition. The total number of indicators, considering both carbon emissions and GDP perspectives, amounts to nine, as illustrated in Figure 1.

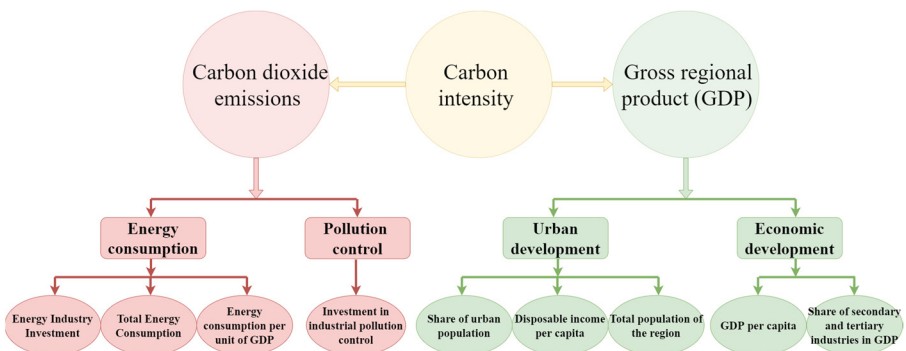

**Figure 1.** Factors influencing the spatial and temporal variability of carbon intensity.

3.1.3. Data Sources

Data on the spatial and temporal variations of carbon intensity were obtained from several authoritative publications: China Statistical Yearbook, China Environmental Statistical Yearbook, China Social Statistical Yearbook, China Population and Employment Statistical Yearbook, and China Energy Statistical Yearbook. To compensate for the shortcomings of these datasets, interpolation was used to supplement the missing information. In addition, the carbon emission data used to calculate carbon intensity were obtained from the provincial inventories of the China Carbon Accounting Databases (CEADs).

*3.2. Research Methodology*

3.2.1. The STIRPAT Model

The STIRPAT model, an advancement over the IPAT equation originally proposed by York, R and Dietz, T addresses the limitations inherent in the IPAT's approach to quantifying human environmental impact [30]. The IPAT equation, which represents the impact of human activities on the environment, considers factors such as population size, affluence, and technological progress. However, its reliance on single and fixed influence factors restricts its applicability. In contrast, the STIRPAT model enhances the IPAT framework by allowing for the inclusion of a broader range of indices in the analysis of influence factors, thereby offering a more flexible and comprehensive approach. This model has become a prevalent method in contemporary environmental research [31–34]. The basic formulation of the STIRPAT model is as follows:

$$I = aP^b A^c T^d e \tag{1}$$

In this model, *I*, *P*, *A*, and *T* represent environmental pressure, population size, affluence, and technology, respectively. The coefficients *b*, *c*, and *d* denote the elasticity of the respective drivers: population size, affluence, and technological progress. The coefficient *a* is a constant that anchors the model, while e represents the error term in the model. To linearize the relationship, logarithms are taken on both sides of the equation, resulting in the following form:

$$\ln I = \ln a + b(\ln P) + c(\ln A) + d(\ln T) + \ln e \tag{2}$$

The model can be extended to:

$$\begin{aligned} \ln C = \ln a &+ B_1(\ln X_1) + B_2(\ln X_2) + B_3(\ln X_3) + B_4(\ln X_4) + B_5(\ln X_5) + \\ &B_6(\ln X_6) + B_7(\ln X_7) + B_8(\ln X_8) + B_9(\ln X_9) + \ln e \end{aligned} \tag{3}$$

where *C* denotes carbon intensity; *a* is the coefficient of the model; $X_1$ is the GDP per capita; $X_2$ is the energy industry investment; $X_3$ is the proportion of secondary and tertiary industries in GDP; $X_4$ is the proportion of urban population; $X_5$ is the disposable income per capita; $X_6$ is the total energy consumption; $X_7$ is the energy consumption per unit of GDP; $X_8$ is the amount of investment in the treatment of industrial pollution; $X_9$ is the total population of the region; *e* is the error of the model; $B_i$ is the elasticity coefficient of each variable, which means that a 1% change in $X_i$ will cause the carbon intensity to change to % will cause the carbon intensity of $B_i$% change.

3.2.2. Geoprobe Model

Spatial differentiation is the spatial manifestation of natural and socioeconomic processes, and probes are statistical methods for detecting spatial variability, as well as revealing the driving factors behind it [35,36]. The causal relationship between independent variables and dependent variables makes both of them have certain spatial distribution

similarities, and detectors are chosen for spatial differentiation factor analysis to reveal the driving force of each influencing factor on the spatial differentiation of carbon intensity [37].

$$q = 1 - \frac{\sum_{h=1}^{L} N_h \sigma_h^2}{N\sigma^2} = 1 - \frac{SSW}{SST} \tag{4}$$

$$SSW = \sum_{h=1}^{L} N_h \sigma_h^2, \; SST = N\sigma^2 \tag{5}$$

where $q$ represents the explanatory power of the factor (its value range is [0, 1]; the closer to 1, the greater the explanatory power); $h$ is the independent variable or dependent variable into (Strata), that is, categorized or partitioned; $N_h$ and $N$ are the number of cells in the layer $h$ and the whole region, respectively; $\sigma_h$ and $\sigma^2$ represent the variance of the independent variable in the layer $h$ and the whole region, respectively; $SSW$ and $SST$ are the sum of Within Sum of Squares (Within Sum of Squares) and the total variance of the whole region (Within Sum of Squares), respectively. Within Sum of Squares.

3.2.3. Temporal and Spatial Development Forecasting Models

Grey system theory is the study of the solution of grey system analysis, modeling, prediction, decision-making and control of the theory [38,39]. This paper adopts the grey system in the GM (1:1) for series prediction [40], as follows:

(1)  Let the original series be $x^{(0)} = \left\{ x^{(0)}(1), x^{(0)}(2) \cdots x^{(0)}(M) \right\}$, and perform one accumulation on $x^{(0)}$ to obtain the new series $x^{(1)} = \left\{ x^{(1)}(1), x^{(1)}(2) \cdots x^{(1)}(M) \right\}$.

(2)  Approximate the differential equation:

$$dx^{(1)}/dt + \alpha x^{(1)} = \mu \tag{6}$$

where $\alpha$ is the developmental gray; $\mu$ is the endogenous control gray.

(3)  Solved by least squares fitting $\alpha, \mu$:

$$\begin{bmatrix} \alpha \\ \mu \end{bmatrix} = (B^T B)^{-1} B^T Y_M \tag{7}$$

(4)  Substituting the required value into the time response function:

$$\hat{x}^{(1)}(k+1) = \left[ x^{(1)}(1) - \frac{\mu}{\alpha} \right] e^{-\alpha t} + \frac{\mu}{\alpha} \tag{8}$$

(5)  Derivative reduction of the above equation yields the predictive model:

$$\hat{x}^{(0)}(k+1) = -\alpha \left[ x^{(0)}(1) - \frac{\mu}{\alpha} \right] e^{-\alpha t} \tag{9}$$

(6)  The gray prediction formula is tested for its accuracy level. If the accuracy test fails, the prediction model needs to be adjusted to obtain the Small Error Probability Test ($p$) and the Variance Ratio Test (C).

## 4. Results and Analysis

### 4.1. Differences in the Spatial and Temporal Distribution of Carbon Intensity

Firstly, carbon intensity within the western region was calculated by comparing carbon emissions to GDP. Temporal variations in carbon intensity were visually depicted using Origin 2021 software, with point and line maps created to illustrate fluctuation trends over time. Furthermore, spatial differences in carbon intensity were illustrated through regional grading utilizing ArcGIS 10.2 software, as demonstrated in Figures 2 and 3.

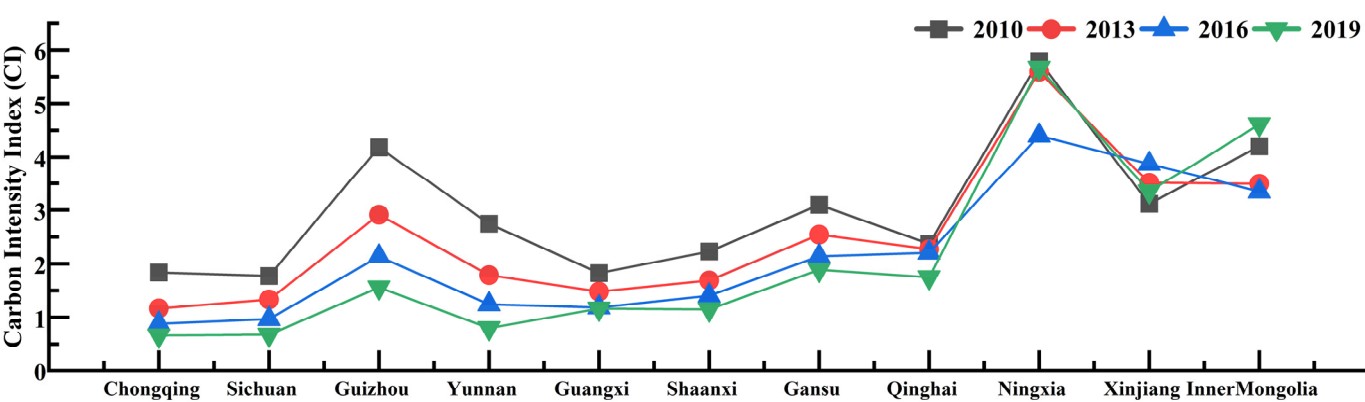

**Figure 2.** Time distribution of carbon intensity (2010–2019).

As depicted in Figure 2, the overall temporal trend of carbon intensity demonstrates a year-on-year decrease. However, specific localities exhibit notable fluctuations, particularly in Ningxia, Xinjiang, and Inner Mongolia.

In Ningxia, the carbon intensity initially experiences a decline, followed by an upward shift. The most significant fluctuation in carbon intensity occurred between 2013 and 2016, with a subsequent decreasing trend. In Xinjiang, there is an initial increase in carbon intensity, succeeded by a decline. The period from 2016 to 2019 witnessed the most substantial fluctuation in carbon intensity, followed by a decreasing trend. Inner Mongolia, conversely, displays an initial decrease in carbon intensity, succeeded by a subsequent rise. The years from 2016 to 2019 witnessed a noticeable increase in carbon intensity fluctuations. In contrast, the remaining western regions consistently exhibited a decreasing trend in carbon intensity year after year.

Over a broad temporal scale, the annual reduction in carbon intensity reflects China's progress in enhancing energy efficiency and transitioning towards a low-carbon economy. This progress is likely associated with a suite of national-level energy-saving and emission-reduction policies, along with technological innovations. Despite the general trend of declining carbon intensity, the trends in Ningxia, Xinjiang, and Inner Mongolia reveal significant inter-regional differences. These discrepancies may be attributed to variations in economic structure, energy configuration, industrial layout, and technological levels across regions [41–44]. Therefore, policy formulation needs to take into account regional economic and energy structure characteristics in order to design low-carbon development strategies that are appropriate to local realities.

Figure 3 illustrates a distinct concentration of carbon intensity in the western region, particularly evident with higher levels in the north-central compared to the southern part. Notably, a consistent year-by-year decline in carbon intensity is observed.

In the year 2010, the Inner Mongolia Autonomous Region, the Ningxia Hui Autonomous Region, and Guizhou Province exhibited notably elevated levels of carbon intensity. During this period, carbon emissions were approximately 5 to 6 times the respective regional GDP. This emphasizes a significant carbon footprint associated with the developmental activities in these regions. The high carbon intensity underscores the intricate relationship between economic development and heightened carbon emissions within these areas. In 2013, a comparison with 2010 revealed relatively stable carbon emissions in the Ningxia Hui Autonomous Region. Despite the stability, carbon intensity remained persistently high. Concurrently, Inner Mongolia and Guizhou experienced a transition from high to medium carbon intensity levels during this period.

Moving to 2016, there was a general decline in carbon intensity across all western regions, except for Xinjiang and Inner Mongolia, where levels remained relatively stable. Notably, in 2019, Inner Mongolia is anticipated to experience a rebound in carbon intensity, while other regions are expected to maintain their existing levels with no significant changes.

This temporal analysis underscores the dynamic nature of carbon emissions, reflecting both stability and fluctuations in different regions over the specified time frame.

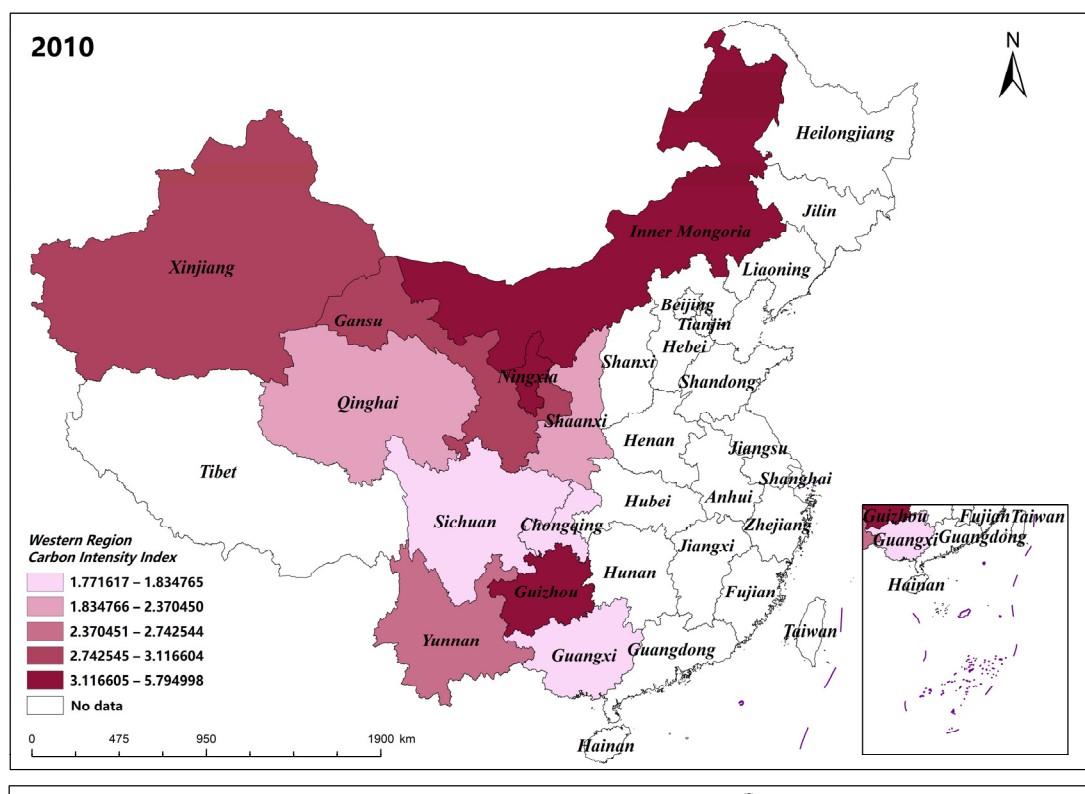

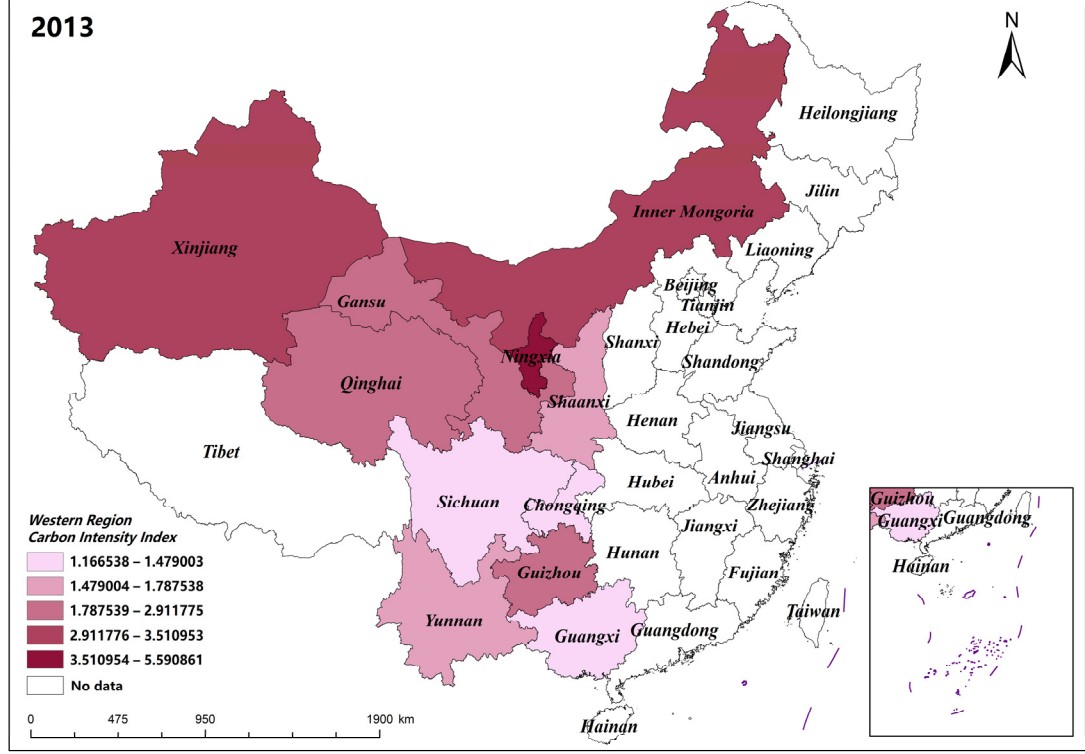

**Figure 3.** *Cont.*

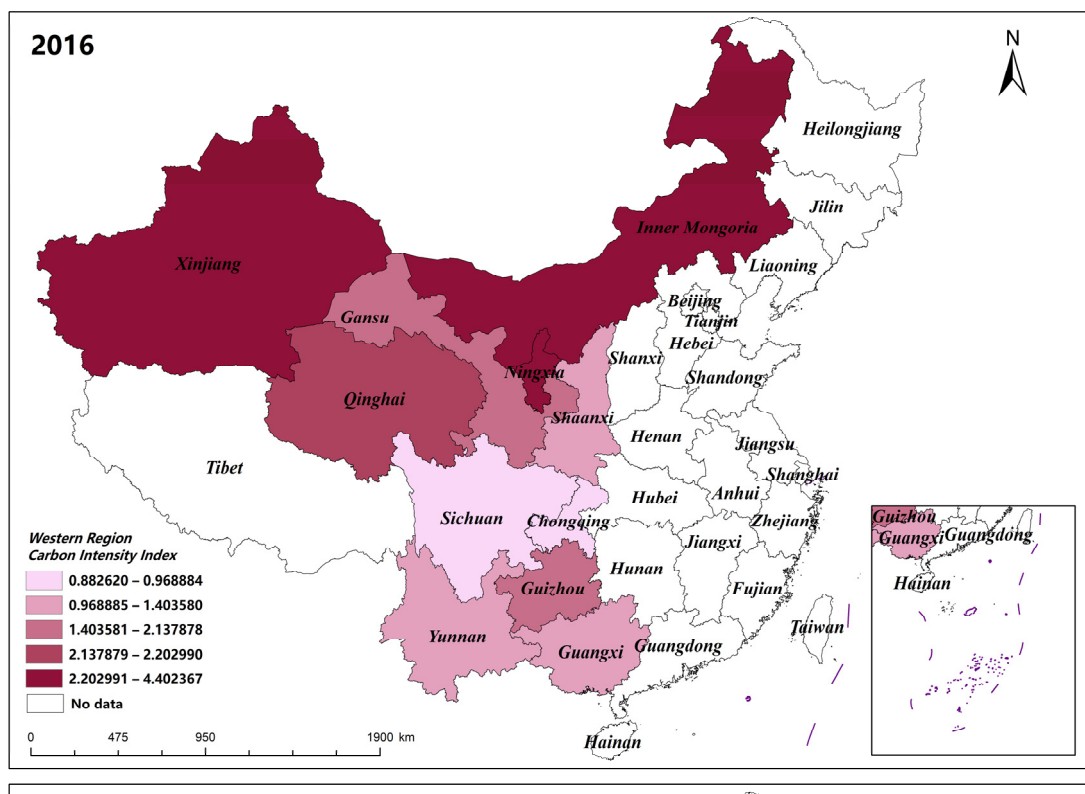

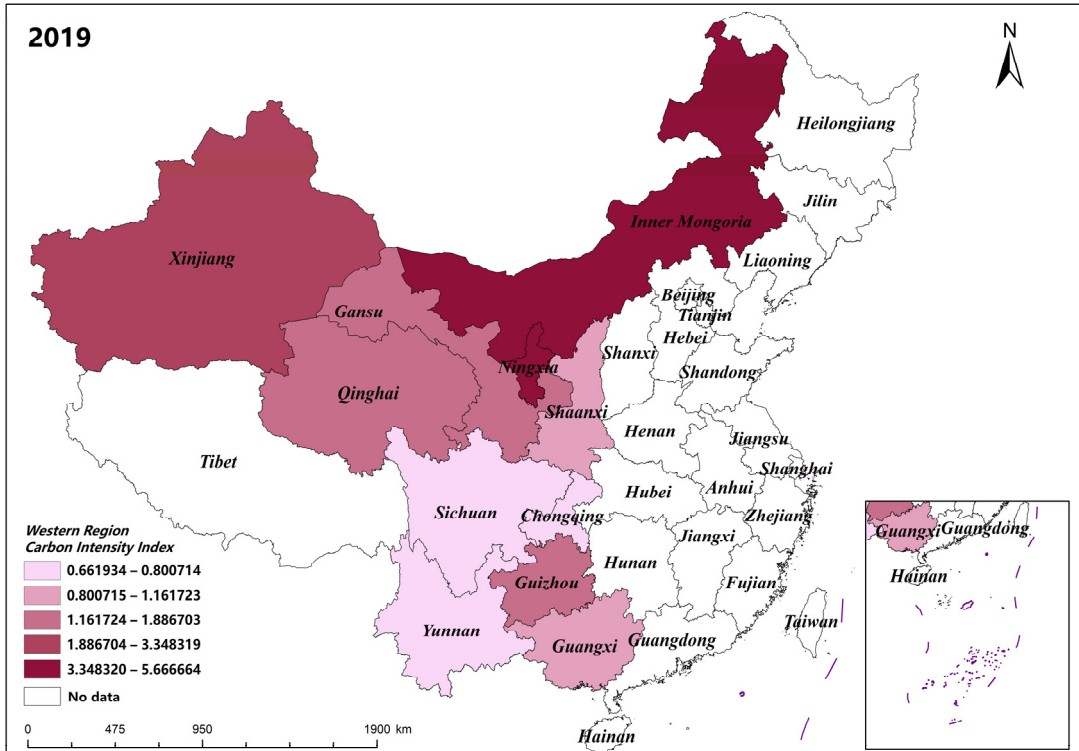

**Figure 3.** Spatial distribution of carbon intensity (2010–2019).

The distribution of carbon intensity exhibits different trends over time and space. This distribution may be attributed to factors such as industrial structure, energy consumption patterns, economic development levels, and geographical and climatic conditions [44,45]. The northern region's reliance on heavy industry and fossil fuels such as coal likely contributes to its elevated carbon intensity. However, the entire western region demonstrates

a year-over-year declining trend in carbon intensity, indicating progress in enhancing energy efficiency and reducing carbon emissions. This progress is likely due to national initiatives for green, low-carbon development, adjustments in the energy structure, and the application of clean energy technologies [46].

### 4.2. Analysis of Temporal Factors Influencing Carbon Intensity

The STIRPAT model was employed to analyze the influencing factors depicted in Figure 1, identifying the determinants of carbon intensity over time in the western regions. The operational procedure of the STIRPAT model begins with the identification of significant factors that impact carbon intensity. This is followed by an analysis of collinearity using ordinary least squares (OLS). Finally, ridge regression analysis is conducted based on the significance analysis and ordinary least squares analysis.

The chosen variables undergo a rigorous scientific selection process, where significant independent variables are selected using the stepwise regression forward and backward method. As depicted in Table 1, the resulting variables GDP per capita, energy industry investment, disposable income per capita, total energy consumption, energy consumption per unit of GDP, and investment in industrial pollution control exhibit potential relationships with the dependent variable. To delve deeper, linear regression was employed to further validate these relationships.

**Table 1.** Description of model variables.

| Name (of a Thing) | Forward Method | Backward Method |
|---|---|---|
| GDP per capita | insignificant | 0.001 |
| Energy industry investment | insignificant | $p < 0.001$ |
| Share of secondary and tertiary industries in GDP | insignificant | insignificant |
| proportion of urban population | insignificant | insignificant |
| Disposable income per capita | insignificant | $p < 0.001$ |
| Total energy consumption | insignificant | 0.003 |
| Energy consumption per unit of GDP | $p < 0.001$ | $p < 0.001$ |
| Investment in industrial pollution control | $p = 0.016$ | $p < 0.001$ |
| Total population | insignificant | insignificant |

Based on the findings of the significance analysis, GDP per capita, energy industry investment, disposable income per capita, total energy consumption, energy consumption per unit of GDP, and investment in industrial pollution control were selected as independent variables, with carbon intensity serving as the dependent variable for analysis using ordinary least squares (OLS). The results are presented in Table 2. The model successfully passed the F-test (F = 517.676, $p = 0.000 < 0.05$), indicating that at least one of GDP per capita, energy industry investment, disposable income per capita, total energy consumption, energy consumption per unit of GDP, and investment in industrial pollution control significantly influences carbon intensity. The study examined the relationships between GDP value, energy industry investment, disposable income per capita, total energy consumption, energy consumption per unit of GDP, and industrial pollution control investment amount, to determine their impact on carbon intensity. Additionally, a multiple covariance test applied to the model identified a VIF (Variance Inflation Factor) value exceeding 10 for energy industry investment, signaling a covariance issue. This indicates that the coefficients obtained from the results of the ordinary least squares fitting cannot be reliably guaranteed, and, consequently, cannot serve as a solid scientific foundation.

**Table 2.** Analysis of ordinary least squares results.

| | Non-Standardized Coefficient | | Standardized Coefficient | t | p | Covariance Diagnosis | |
|---|---|---|---|---|---|---|---|
| | B | Standard Error | Beta | | | VIF | Tolerance |
| A constant (math.) | −2.178 | 2.728 | - | −0.798 | 0.469 | - | - |
| GDP per capita | 1.158 | 0.118 | 0.237 | 9.813 | 0.001 | 1.820 | 0.550 |
| energy industry investment | −1.623 | 0.130 | −0.710 | −12.475 | 0.000 | 10.067 | 0.099 |
| disposable income per capita | −2.313 | 0.249 | −0.254 | −9.288 | 0.001 | 2.332 | 0.429 |
| total energy consumption | 0.812 | 0.131 | 0.302 | 6.191 | 0.003 | 7.415 | 0.135 |
| energy consumption per unit of GDP | 2.960 | 0.070 | 1.019 | 42.502 | 0.000 | 1.788 | 0.559 |
| investment in industrial pollution control | 1.381 | 0.088 | 0.656 | 15.609 | 0.000 | 5.501 | 0.182 |
| $R^2$ | 0.999 | | | | | | |
| Adjustment $R^2$ | 0.997 | | | | | | |
| F | F (6,4) = 517.676, $p$ = 0.000 | | | | | | |
| D-W value | 2.531 | | | | | | |

Note: Dependent variable is carbon intensity; $p < 0.05$; $p < 0.01$.

To mitigate the interference resulting from the multiple covariances inherent in panel data, and to preserve a more substantial amount of information from both the independent variables and the dependent variable, the ridge regression method was adopted for data analysis. This approach addresses the challenges posed by multiple covariance interference in panel data analysis while maximizing the retention of critical information from the independent and dependent variables. To effectively address the issue of multiple covariance interference in panel data and better preserve the information contained in the independent and dependent variables, the data were reanalyzed using the ridge regression method.

The ridge regression program was implemented using SPSS 25.0 software. The analysis involved equations, ridge trace plots, and goodness-of-fit assessments across various values of the ridge parameter (k). The independent variables included per capita GDP, energy industry investment, per capita disposable income of residents, total energy consumption, energy consumption per unit of GDP, and investment in the treatment of industrial pollution. Carbon intensity was the designated dependent variable.

Upon setting k = 0.05, the ridge coefficients exhibited stabilization, and the model's R-squared value reached 0.983. This indicates that per capita GDP, energy industry investment, per capita disposable income, total energy consumption, energy consumption per unit of GDP, and investment in industrial pollution control collectively explain 98.3% of the variance in carbon intensity. Detailed results of the ridge regression at k = 0.05 are presented in Table 3. Furthermore, the model underwent an F-test, yielding a statistically significant result (F = 37.766, $p$ = 0.002 < 0.05). This implies that at least one of the variables, namely per capita GDP, energy industry investment, disposable income per capita, total energy consumption, energy consumption per unit of GDP, or industrial pollution control investment, significantly influences the relationship with carbon intensity. The non-standard coefficient equation derived from ridge regression aligns with the STIRPAT model equation.

$$\ln C = 7.031 + 1.256(\ln X_1) - 0.810(\ln X_2) - 2.753(\ln X_5) + 0.172(\ln X_6) + 2.611(\ln X_7) + 0.956(\ln X_8) \tag{10}$$

where $C$ denotes carbon intensity; $X_1$ is GDP per capita, $X_2$ is energy industry investment, $X_5$ is disposable income per capita, $X_6$ is total energy consumption, $X_7$ is energy consumption per unit of GDP, and $X_8$ is investment in industrial pollution control.

**Table 3.** Results of ridge regression at k = 0.05.

| | Non-Standardized Coefficient | | Standardized Coefficient | t | $p$ | VIF Value |
|---|---|---|---|---|---|---|
| | **B** | **Standard Error** | **Beta** | | | |
| A constant (math.) | 7.031 | 7.507 | - | 0.937 | 0.402 | - |
| GDP per capita | 1.256 | 0.372 | 0.257 | 3.378 | 0.028 | 1.339 |
| energy industry investment | −0.810 | 0.260 | −0.354 | −3.120 | 0.036 | 2.971 |
| disposable income per capita | −2.753 | 0.722 | −0.303 | −3.813 | 0.019 | 1.453 |
| total energy consumption | 0.172 | 0.296 | 0.064 | 0.582 | 0.592 | 2.792 |
| energy consumption per unit of GDP | 2.611 | 0.210 | 0.899 | 12.409 | 0.000 | 1.210 |
| investment in industrial pollution control | 0.956 | 0.219 | 0.454 | 4.360 | 0.012 | 2.505 |
| $R^2$ | 0.983 | | | | | |
| Adjustment $R^2$ | 0.957 | | | | | |
| F | F (6,4) = 37.766, $p$ = 0.002 | | | | | |

Note: Dependent variable is carbon intensity; $p < 0.05$; $p < 0.01$.

The regression analysis reveals noteworthy findings regarding the impact of various factors on carbon intensity. Specifically: The regression coefficient for per capita GDP is 1.256 (t = 3.378, $p$ = 0.028 < 0.05), indicating a significant positive influence on carbon intensity. Energy industry investment is associated with a regression coefficient of −0.810 (t = −3.120, $p$ = 0.036 < 0.05), signifying a significant negative impact on carbon intensity. Per capita disposable income exhibits a regression coefficient of −2.753 (t = −3.813, $p$ = 0.019 < 0.05), suggesting a significant negative effect on carbon intensity. Total energy consumption's regression coefficient is 0.172 (t = 0.582, $p$ = 0.592 > 0.05), implying a non-significant positive impact on carbon intensity. The regression coefficient for energy consumption per unit of GDP is 2.611 (t = 12.409, $p$ = 0.000 < 0.01), indicating a significant positive influence on carbon intensity. Industrial pollution control investment is associated with a regression coefficient of 0.956 (t = 4.360, $p$ = 0.012 < 0.05), signifying a significant positive impact on carbon intensity.

Results derived from the STIRPAT model indicate that per capita GDP, energy consumption per unit of GDP, and investment in industrial pollution control significantly contribute to the increase in carbon intensity. Conversely, investment in the energy sector and per capita disposable income are associated with a significant reduction in carbon intensity. In this analysis, total energy consumption does not have a statistically significant impact on carbon intensity. Per capita GDP's rise is tied to higher carbon intensity, suggesting economic growth's environmental toll. Clean energy investments are inversely related to carbon intensity, indicating efficiency gains. Higher incomes align with greener consumption, aiding emission cuts. Energy use per GDP unit reflects efficiency deficits in high-intensity economies. Despite pollution control investments suggesting increased industrial activity, a holistic strategy promoting sustainable growth, clean energy, and low-carbon habits is key to reducing carbon intensity, with policy evaluations needing to account for complex economic-social dynamics.

### 4.3. Analysis of Factors Affecting Carbon Intensity in the Spatial Dimension

The Geo-detector model was utilized to analyze the influencing factors illustrated in Figure 1, precisely identifying the determinants of carbon intensity within the spatial dimension of the western regions. Initially, the reclassification tool in ArcGIS was employed to convert the original continuous data raster. Subsequent calculations were performed in Excel 2016, with the Geo-detector 2015 software acting as a macro within the Excel spreadsheet to directly process the data for analysis. The outcomes included both univariate and bivariate analyses. The results of the univariate and bivariate analyses are detailed in Tables 4 and 5.

**Table 4.** Explanatory power of factors influencing carbon intensity in the western region.

| Impact Level | Detection Indicators | Explanatory Power (q-Value) | | | | |
|---|---|---|---|---|---|---|
| | | **2010** | **2013** | **2016** | **2019** | **Synthesize** |
| Energy Consumption | Investment in the energy industry | 0.178 | 0.254 | 0.230 | 0.172 | 0.209 |
| | Total energy consumption | 0.192 | 0.320 | 0.526 | 0.364 | 0.351 |
| | Energy consumption per unit of GDP | 0.814 | 0.636 | 0.735 | 0.834 | 0.755 |
| Pollution Control | Investment in industrial pollution control | 0.304 | 0.190 | 0.248 | 0.105 | 0.212 |
| Urban Development | Share of urban population | 0.171 | 0.113 | 0.100 | 0.446 | 0.208 |
| | Per capita disposable income | 0.042 | 0.847 | 0.340 | 0.512 | 0.435 |
| | Total population of the region | 0.368 | 0.458 | 0.464 | 0.386 | 0.419 |
| Economic Development | GDP per capita | 0.319 | 0.105 | 0.120 | 0.221 | 0.191 |
| | Share of secondary and tertiary industries in GDP | 0.113 | 0.120 | 0.087 | 0.059 | 0.095 |

From the results of the Single-factor analysis of variance (Table 4), it is evident that energy consumption per unit of GDP and the total regional population exerts a more substantial explanatory influence on the spatial variation of carbon intensity in the western region. Particularly noteworthy is the average explanatory power of energy consumption per unit of GDP, which attains 0.755, signifying the highest degree of impact on the spatial differentiation of carbon intensity. Conversely, the category with the least explanatory power regarding the spatial distribution differences in carbon intensity is the proportion of secondary and tertiary industries to GDP, registering an average explanatory power of 0.095. Assessing the level of influence, it is discerned that energy consumption and urban development wield a more pronounced effect on carbon intensity.

The results of the two-factor analysis of variance (Table 5) reveal that the impact of two-factor interaction on the spatial differentiation of carbon intensity in the western region surpasses that of a single factor, demonstrating non-linear enhancement and two-factor synergies. In 2010, the most robust interaction effect occurred between the proportion of urban population and energy consumption per unit of GDP, yielding an explanatory power of 0.996. This underscores that changes in the urban population proportion and energy consumption per unit of GDP were the primary driving forces behind the spatial variation in carbon intensity. In 2013, the interaction between total regional population and GDP per capita exhibited the highest level of influence, boasting an explanatory power of 0.996. This suggests that the total regional population and GDP per capita played pivotal roles in shaping the spatial distribution of carbon intensity. Similar to 2013, 2016 witnessed the most potent interaction between the total regional population and GDP per capita, registering an explanatory power of 1.000. The paramount two-factor explanatory power in 2019 was associated with the total regional population and urban population share, with an explanatory power of 0.998.

**Table 5.** Interactive detection results of carbon intensity in the western region.

| Particular Year | | $X_1$ | $X_2$ | $X_3$ | $X_4$ | $X_5$ | $X_6$ | $X_7$ | $X_8$ | $X_9$ |
|---|---|---|---|---|---|---|---|---|---|---|
| | $X_1$ | 0.319 | | | | | | | | |
| | $X_2$ | **0.519** | 0.178 | | | | | | | |
| | $X_3$ | **0.394** | 0.471 | 0.113 | | | | | | |
| | $X_4$ | **0.574** | 0.612 | **0.312** | 0.171 | | | | | |
| **2010** | $X_5$ | 0.638 | 0.901 | 0.451 | 0.478 | 0.042 | | | | |
| | $X_6$ | 0.892 | **0.434** | 0.603 | 0.560 | 0.992 | 0.192 | | | |
| | $X_7$ | **0.992** | **0.897** | **0.992** | **0.996** | **0.901** | **0.990** | 0.814 | | |
| | $X_8$ | 0.894 | **0.406** | **0.499** | **0.542** | 0.973 | **0.446** | **0.961** | 0.304 | |
| | $X_9$ | 0.920 | **0.510** | **0.474** | **0.474** | 0.833 | **0.485** | **0.950** | **0.451** | 0.368 |

| Particular Year | | $X_1$ | $X_2$ | $X_3$ | $X_4$ | $X_5$ | $X_6$ | $X_7$ | $X_8$ | $X_9$ |
|---|---|---|---|---|---|---|---|---|---|---|
| | $X_1$ | 0.105 | | | | | | | | |
| | $X_2$ | **0.449** | 0.254 | | | | | | | |
| | $X_3$ | **0.367** | **0.481** | 0.110 | | | | | | |
| | $X_4$ | 0.403 | 0.692 | 0.878 | 0.113 | | | | | |
| **2013** | $X_5$ | **0.989** | **0.983** | **0.994** | **0.996** | 0.847 | | | | |
| | $X_6$ | 0.965 | **0.600** | 0.994 | 0.672 | **0.986** | 0.320 | | | |
| | $X_7$ | 0.989 | **0.701** | 0.994 | **0.703** | **0.983** | **0.693** | 0.636 | | |
| | $X_8$ | 0.808 | **0.490** | 0.800 | 0.808 | **0.876** | 0.930 | 0.991 | 0.190 | |
| | $X_9$ | 0.996 | **0.626** | 0.994 | **0.560** | **0.994** | **0.615** | **0.680** | 0.972 | 0.459 |

| Particular Year | | $X_1$ | $X_2$ | $X_3$ | $X_4$ | $X_5$ | $X_6$ | $X_7$ | $X_8$ | $X_9$ |
|---|---|---|---|---|---|---|---|---|---|---|
| | $X_1$ | 0.120 | | | | | | | | |
| | $X_2$ | **0.360** | 0.230 | | | | | | | |
| | $X_3$ | 0.372 | 0.619 | 0.087 | | | | | | |
| | $X_4$ | **0.367** | 0.741 | 0.754 | 0.099 | | | | | |
| **2016** | $X_5$ | 0.670 | **0.493** | 0.710 | 0.678 | 0.340 | | | | |
| | $X_6$ | 0.967 | **0.748** | 0.965 | 0.976 | **0.842** | 0.526 | | | |
| | $X_7$ | **0.880** | 0.845 | 0.976 | **0.889** | **0.841** | **0.819** | 0.735 | | |
| | $X_8$ | **0.375** | **0.693** | 0.668 | **0.375** | **0.651** | 0.952 | **0.913** | 0.248 | |
| | $X_9$ | **1.000** | **0.787** | 0.907 | 0.814 | **0.575** | **0.802** | **0.805** | 0.859 | 0.464 |

| Particular Year | | $X_1$ | $X_2$ | $X_3$ | $X_4$ | $X_5$ | $X_6$ | $X_7$ | $X_8$ | $X_9$ |
|---|---|---|---|---|---|---|---|---|---|---|
| | $X_1$ | 0.221 | | | | | | | | |
| | $X_2$ | 0.679 | 0.172 | | | | | | | |
| | $X_3$ | 0.987 | **0.593** | 0.059 | | | | | | |
| | $X_4$ | **0.681** | **0.681** | **0.638** | 0.446 | | | | | |
| **2019** | $X_5$ | 0.905 | **0.994** | 0.945 | **0.919** | 0.512 | | | | |
| | $X_6$ | 0.878 | **0.445** | **0.593** | **0.878** | **0.751** | 0.364 | | | |
| | $X_7$ | **0.987** | **0.991** | **0.856** | **0.946** | **0.952** | **0.991** | 0.834 | | |
| | $X_8$ | 0.681 | **0.249** | 0.593 | **0.681** | **0.619** | **0.422** | **0.987** | 0.105 | |
| | $X_9$ | 0.750 | **0.974** | 0.942 | 0.998 | **0.703** | **0.729** | **0.959** | 0.732 | 0.386 |

Note: Single underlined data are nonlinear enhancements, bolded and boldface data are two-factor enhancements.

In summary, the explanatory power of two-factor interactions during the period 2010–2019 spans from the proportion of urban population and energy consumption per unit of GDP to the total regional population and GDP per capita, and ultimately to the total regional population and the proportion of urban population. The dynamic evolution of interaction explanatory power suggests that the primary influence on spatial variation in carbon intensity stems from the interaction between energy consumption and urban development.

The rationale behind the observed spatial variance in carbon intensity being impacted by energy consumption and urban development may lie in the fact that energy serves as a fundamental indicator of economic development. Simultaneously, energy consumption directly results in carbon emissions, influencing both GDP values and carbon emissions, thereby affecting carbon intensity. Moreover, cities and towns represent focal points of human activities, and their developmental levels directly impact regional economies. The processes associated with human activities contribute to carbon emissions, further

influencing carbon intensity. Consequently, the interaction between energy consumption and urban development, as core factors influencing carbon intensity, emerges as the central explanatory power for spatial disparities in carbon intensity distribution.

Results obtained using the Geo-detector model reveal that in the univariate analysis, energy consumption per unit of GDP and Disposable income per capita exert the strongest influence on the spatial variation of carbon intensity. Bivariate interaction analysis discloses that the proportion of the proportion of the urban population and energy consumption per unit of GDP, as well as the Total population and per capita GDP, have the most pronounced impact on the spatial variation of carbon intensity at different time intervals.

### 4.4. Projections of Spatial and Temporal Development of Carbon Intensity

This study employs raw data related to carbon intensity in the western regions from 2010 to 2021 and utilizes the GM (1:1) grey prediction model to forecast the development of carbon intensity. With the assistance of MATLAB 2019a software, Equations (6)–(9) are applied to process the raw data, yielding predicted values for the carbon intensity development in the western regions. Precision testing and error analysis of the development forecast are conducted (see Table 6 for detailed criteria), resulting in a Small Error Probability Test ($p$) value of 1, and Variance Ratio Test (C) values of 0.032, 0.028, 0.014, 0.028, 0.050, 0.025, 0.017, 0.086, 0.183, 0.155, and 0.157, respectively. Through rigorous accuracy testing and error analysis, the prediction model demonstrates high precision.

**Table 6.** $p$ and C value accuracy prediction level.

| Predictive Accuracy (Name) | Excellent | Qualified | Medium | Unqualified |
|---|---|---|---|---|
| Variance Ratio Test (C) | >0.95 | >0.80 | >0.70 | $\leq$0.70 |
| Small Error Probability Test ($p$) | $\leq$0.35 | $\leq$0.50 | $\leq$0.65 | $\geq$0.65 |

As evident from Figure 4, the actual average carbon intensity for the western regions in 2021 was 2.14. This indicates that the total carbon emissions were 2.14 times the Gross Domestic Product (GDP) of the region, highlighting the correlation between regional economic development and an increase in carbon emissions. The outcomes of the prediction model reveal a yearly decreasing trend in regional average carbon intensity. The predicted average carbon intensity for 2029 stands at 1.576, suggesting a declining ratio between the economy and the environment. This implies a reduction in environmental loss during economic development and an enhancement in environmental protection benefits throughout the economic construction process.

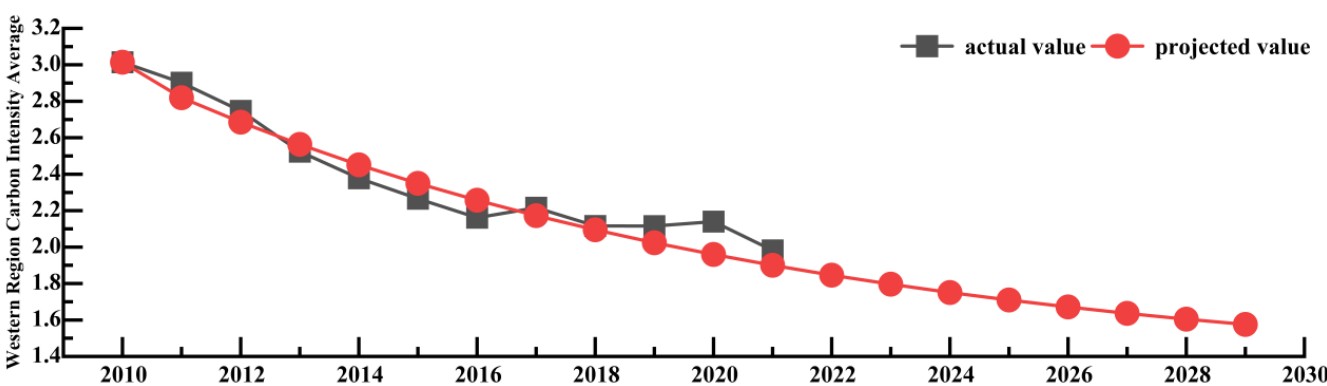

**Figure 4.** Comparison of actual and projected carbon intensity development in the western region.

Building on the analysis of factors influencing the spatio temporal distribution of carbon intensity, the current trend in the average actual carbon intensity in the western regions is affected by energy consumption per unit of GDP, per capita GDP, investment in industrial pollution control, investment in the energy sector, per capita disposable income,

and the urban population ratio. Model predictions suggest that from 2021 to 2029, the average carbon intensity in the western regions is expected to decrease annually by 7.01%. Compared to the national target set in the "14th Five-Year Plan" of reducing carbon intensity by 65% by 2030 relative to 2005 levels, the current trajectory appears promising. To maintain this positive trend in carbon intensity, it is imperative to address and reinforce the temporal and spatial factors influencing it. Policy recommendations should stem from these factors, including energy consumption per unit of GDP, per capita GDP, investment in industrial pollution control, investment in the energy sector, per capita disposable income, and the urban population ratio.

## 5. Conclusions

The spatial and temporal variations in carbon intensity within the western region are pronounced. Temporally, there has been a general downward trend in carbon intensity over the years, albeit with localized fluctuations. Specifically, regions like Ningxia and Inner Mongolia exhibit a pattern where carbon intensity first decreases and then increases. Conversely, in Xinjiang, the trend in carbon intensity initially increases before decreasing. Spatially, areas with a high carbon intensity ratio are predominantly located in the northern part of the western region, with Ningxia, Inner Mongolia, and Xinjiang continuing to be regions with a high ratio.

In this study, we employ the STIRPAT model to examine the trend of changes over time and utilize stepwise regression to screen nine categories of influencing factors, ultimately identifying significant factors such as per capita GDP, investment in the energy sector, per capita disposable income, total energy consumption, energy consumption per unit of GDP, and investment in industrial pollution control. Further analysis through ridge regression reveals that, in the temporal dimension, per capita GDP, energy consumption per unit of GDP, and investment in industrial pollution control significantly positively impact carbon intensity. Conversely, investment in the energy industry and per capita disposable income exhibit a significant negative effect on carbon intensity.

This study employs geographic probes to analyze the spatial patterns and factor interactions affecting carbon intensity. Single-factor analysis reveals that the northern region's high carbon intensity primarily stems from its energy consumption per GDP unit and population size. Interaction analysis shows varying significant two-factor interactions over time. In 2010, urban population and energy consumption per GDP unit had the highest explanatory power (0.996), shifting in 2013 to total population and per capita GDP, maintaining the same explanatory power. By 2019, the most explanatory interaction was between the total population and urban population proportion, with an explanatory power of 0.998.

Finally, we forecast the spatio-temporal evolution of the average carbon intensity in the western region, achieving a high degree of fit and prediction accuracy. The results indicate a decreasing trend in the carbon intensity ratio, suggesting that harmonizing economic and environmental development represents the future trajectory of progress. In the endeavor to balance economic and environmental objectives, it is crucial to consider the variations in spatial and temporal distributions and to engage in multi-regional coordination to mitigate emissions and foster development.

## 6. Policy Proposals

Upon analyzing the outcomes and key conclusions, it is determined that the principal elements influencing the spatial and temporal fluctuations of carbon emission intensity include energy consumption per unit of GDP, GDP per capita, investment in industrial pollution control, investment in the energy sector, per capita disposable income and the urban population ratio. In light of these findings, the subsequent policy recommendations are proposed:

Firstly, it is essential to implement comprehensive strategies to enhance mechanisms for coordinated regional economic development and tackle the disparities in carbon inten-

sity's spatial and temporal distribution. Initially, creating a robust regional cooperation framework involving government agencies, businesses, and research institutions will support information exchange and collective decision-making. Secondly, adopting policies and incentives tailored to the specific conditions of each region can efficiently reduce variations in carbon intensity. Furthermore, investments in renewable energy infrastructure, the enhancement of energy efficiency standards, the promotion of green technologies, and the encouragement of public awareness and participation through educational and outreach programs are pivotal. These actions will facilitate a more equitable and sustainable development trajectory, simultaneously addressing the spatial and temporal differences in carbon intensity distribution effectively.

Secondly, enhancing investment in industrial pollution control and optimizing investment in the energy sector are critical steps towards mitigating the spatial and temporal disparities in carbon intensity. Initially, bolster financial input for the research, development, and dissemination of environmental protection technologies, thereby elevating the level of environmental safeguarding in industrial production. Secondly, institute effective regulatory frameworks to ensure that corporations comply with environmental laws and standards during their operational processes, which in turn will lower carbon emissions. Moreover, by advocating for and aiding businesses in the adoption of clean energy and low-carbon technologies, the adverse effects of the energy sector on carbon intensity can be minimized. Addressing the spatial and temporal variations in carbon intensity can be achieved more effectively through the application of region-specific industrial pollution control policies. Collectively, ensuring prudent investment alongside the execution of all-encompassing environmental measures will facilitate the harmonious development of carbon intensity across various regions.

Thirdly, strategically managing the urban population ratio and elevating residents' per capita disposable income represent crucial strategies for mitigating the spatial and temporal disparities in carbon intensity. Firstly, optimizing urban and rural development policies to foster population mobility can alleviate the population density in areas with high carbon intensity, thereby reducing carbon emissions. Secondly, enhancing residents' per capita disposable income encourages behaviors conducive to low-carbon consumption, diminishes activities associated with high carbon emissions, and ultimately reduces carbon intensity levels. Furthermore, by bolstering urban planning and construction efforts to improve the quality of life and environmental sustainability in urban areas, it is possible to attract migration from rural areas, contributing to reduced carbon emissions in those regions. A holistic approach to understanding the influence of population restructuring and income growth on carbon intensity can significantly ameliorate the spatial and temporal variations in carbon intensity, fostering harmonious economic and environmental development.

**Author Contributions:** Conceptualization, M.Y. and L.W.; methodology, M.Y.; software, M.Y.; validation, M.Y., L.W. and H.H.; formal analysis, M.Y.; investigation, M.Y.; resources, L.W.; data curation, M.Y.; writing—original draft preparation, M.Y.; writing—review and editing, M.Y. and L.W.; visualization, M.Y.; supervision, L.W.; project administration, L.W.; funding acquisition, L.W. All authors have read and agreed to the published version of the manuscript.

**Funding:** This research was funded by the Xinjiang Uygur Autonomous Region Postgraduate Education Teaching Reform Research Project (XJ2022GY14); and the Research Initiation Program for High-level Talents at Shihezi University (RCZK2018C21).

**Institutional Review Board Statement:** Not applicable.

**Informed Consent Statement:** Not applicable.

**Data Availability Statement:** Data are contained within the article.

**Conflicts of Interest:** The authors declare no conflict of interest.

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
