# Peer review of "Analysis of Factors Influencing the Spatial and Temporal Variability of Carbon Intensity in Western China"

_sustainability, doi:10.3390/su16083364_

Round 1

Reviewer 1 Report

Comments and Suggestions for Authors

GENERAL COMMENTS

The abstract has acronyms that are difficult to understand and a list. It needs to be revised for a non-technical reader can understand it.

The introduction can be also revised. Maybe the technical part of the abstract can be included in the introduction including definitons of the key terms and acronyms, because as it is it does not inform the reader what to expect.

The authors are not critical about the predicting power of the indicators (they argue that they have boosted the explanatory power and achieved a high degree of fit and prediction accuracy). But the data for 2022 might change the fit and prediction accuracy and there is also option to design a future that is much better than the predicted (the debate on forecasting vs. backcasting). A discussion section is needed where the policy proposals are described and scrutinized. The conclusions should draw on these discussions. 

IN-TEXT COMMENTS

The country has made commitments to peak carbon dioxide emissions by 2030 45

and achieve carbon neutrality by 2060 in alignment with the global response to climate 46

change. REFERENCE MISSING

BAD FORMATING IN THE FORMULA ON PAGE 10 ln 7.031 1.256(ln ) 0.810(ln ) 2.753(ln )

0.172(ln ) 2.611(ln ) 0.956(ln )

 (10) 356

Sustainability 2024, 16, x FOR PEER REVIEW 12 of 18

Where:C denotes carbon intensity; X1

 is GDP per capita, X2

 is energy industry invest- 357

ment, X5

 is disposable income per capita, X6

 is total energy consumption, X7

 is en- 358

ergy consumption per unit of GDP, and X8

 is investment in industrial pollution control. 3

Reviewer 2 Report

Comments and Suggestions for Authors

The contribution of analysis of factors influencing the spatial and temporal variability of carbon intensity in western China presents a valuable contribution to understanding the key indicators and drivers of the pollution effect of human-related activities. The model reveals that investments in the energy industry and a surplus of disposable income have an influence on energy intensity. Moreover, urban development has a strong influence on total carbon emissions and the spatial distribution of carbon intensity. This is important not just for achieving climate neutrality, but also for city planning and well-being of citizens affected by this influence.

Overall, the presented results provide an interesting insight on influence of factors which are traditionally not directly correlated with energy intensity and emissions. As stated in the essay, enhancing the coordinated development mechanisms of regional economies, boosting investment in industrial pollution control, managing investment in the energy sector, adjusting the urban population ratio, and elevating residents' per capita disposable income8 provide policy mechanisms and goals which can significantly boost the process of achieving carbon neutrality.

General comments:

1. In the abstract part, the results need to be more clearly presented, providing at least the upper and lower values to the most significant factors.

2. The introduction part lacks more references to the existing text, which would be helpful in the first part of the introduction.

Reviewer 3 Report

Comments and Suggestions for Authors

Dear Authors,

This paper analyzes western China's spatiotemporal variability of carbon dioxide emission intensity. The topic is engaging. However, some critical points need to be addressed:

Lack of clarity and coherence: The text needs coherence and clarity in presenting the analysis. It jumps between different methodologies and results without a clear structure.

Inconsistencies in footnotes: The text references various studies but needs to provide appropriate citations or references in the text, which reduces the credibility of the analysis.

Lack of specificity: The text needs to adequately explain them to use technical terms and models, which may confuse readers who are not experts in the field.

Unclear methodology: although the text mentions the use of the STIRPAT and GeoDetector models, it needs to clearly explain how these models were applied or the specific results obtained.

Missing data: The text mentions the exclusion of data from Tibet but needs to provide a clear rationale for this decision or its impact on the overall analysis.

Lack of Discussion: The text presents findings and recommendations but needs a more thorough discussion of the implications of these results or how they contribute to existing knowledge in the field.

The text requires revision to provide a more coherent structure, clarify methodology and results, and ensure proper reference to sources. Furthermore, more in-depth analysis and discussion are needed to support the presented findings.

Additionally, there are sentences in the text that have yet to be translated into English (lines 107-113).

The test is incomplete and has not been prepared correctly for editing.  Therefore, it is not suitable for publication in its current form.

Round 2

Reviewer 3 Report

Comments and Suggestions for Authors

Dear Authors,

Thank you for explaining and taking into account my comments.

In my opinion, the article is suitable for publication in its current form after corrections.